# Extracellular Vesicles from *Bothrops jararaca* Venom Are Diverse in Structure and Protein Composition and Interact with Mammalian Cells

**DOI:** 10.3390/toxins14110806

**Published:** 2022-11-19

**Authors:** Larissa Gonçalves-Machado, Brunno Renato Farias Verçoza, Fábio César Sousa Nogueira, Rafael Donadélli Melani, Gilberto Barbosa Domont, Silas Pessini Rodrigues, Juliany Cola Fernandes Rodrigues, Russolina Benedeta Zingali

**Affiliations:** 1Laboratório de Hemostase e Venenos, Instituto de Bioquímica Médica Leopoldo de Meis (IBqM), Instituto Nacional de Ciência e Tecnologia de Biologia Estrutural e Bioimagem (Inbeb), Universidade Federal do Rio de Janeiro, Rio de Janeiro 21941-902, Brazil; 2Instituto Vital Brazil, Gerência de Desenvolvimento Tecnológico, Niterói 24230-410, Brazil; 3Núcleo Multidisciplinar de Pesquisa em Biologia (NUMPEX-Bio), Universidade Federal do Rio de Janeiro, Campus UFRJ Duque de Caxias, Duque de Caxias, Rio de Janeiro 25240-005, Brazil; 4Laboratório de Química de Proteínas, Unidade Proteômica, Instituto de Química, Universidade Federal do Rio de Janeiro, Rio de Janeiro 21941-909, Brazil; 5Laboratório de Proteômica (LabProt)—LADETEC, Instituto de Química, Universidade Federal do Rio de Janeiro, Rio de Janeiro 21941-598, Brazil

**Keywords:** snake venom, extracellular vesicles, 5′-nucleotidase

## Abstract

Snake venoms are complex cocktails of non-toxic and toxic molecules that work synergistically for the envenoming outcome. Alongside the immediate consequences, chronic manifestations and long-term sequelae can occur. Recently, extracellular vesicles (EVs) were found in snake venom. EVs mediate cellular communication through long distances, delivering proteins and nucleic acids that modulate the recipient cell’s function. However, the biological roles of snake venom EVs, including possible cross-organism communication, are still unknown. This knowledge may expand the understanding of envenoming mechanisms. In the present study, we isolated and characterized the EVs from *Bothrops jararaca* venom (Bj-EVs), giving insights into their biological roles. Fresh venom was submitted to differential centrifugation, resulting in two EV populations with typical morphology and size range. Several conserved EV markers and a subset of venom related EV markers, represented mainly by processing enzymes, were identified by proteomic analysis. The most abundant protein family observed in Bj-EVs was 5’-nucleotidase, known to be immunosuppressive and a low abundant and ubiquitous toxin in snake venoms. Additionally, we demonstrated that mammalian cells efficiently internalize Bj-EVs. The commercial antibothropic antivenom partially recognizes Bj-EVs and inhibits cellular EV uptake. Based on the proteomic results and the in vitro interaction assays using macrophages and muscle cells, we propose that Bj-EVs may be involved not only in venom production and processing but also in host immune modulation and long-term effects of envenoming.

## 1. Introduction

Animal venoms are among the most complex biological fluids produced by several diversified multicellular organisms, including snakes, cone snails, spiders, scorpions, platypus, etc. Snake venoms are mainly composed of soluble proteins and peptides secreted by exocytosis, and other minor, small organic and inorganic molecules [1,2,3]. In terms of protein diversity, venoms undergo evolutionary pressure which generates a rapid response, causing high variability in their toxic components [4,5]. As a result, a large number of different molecules can be found in venoms from the same or different species, demonstrating their complexity. Paradoxically, numerous snake venom toxins belong to a restricted number of protein families, and same-family toxins share strict structural scaffolds, although they can be involved in distinct biological activities [5,6,7,8].

*Bothrops jararaca* (Serpentes, Viperidae) is often involved in envenoming accidents in Brazil. Therefore, both the snake and its venom have been extensively studied [9,10,11,12,13]. *B. jararaca* venom is composed of the main protein families: snake venom metalloproteases (SVMPs), bradykinin-potentiating and C-type natriuretic peptides (BPP-CNP), C-type lectins-like (CTLs), snake venom serine proteases (SVSP), L-amino acid oxidase (LAO), phospholipases A2 (PLA2), snake venom vascular endothelial growth factor (svVEGF), and cysteine-rich secretory protein (CRISP) [10,11,12,13,14]. All these toxins shape clinical manifestations and symptoms of snakebites.

In general, venoms from viperid snakes affect the hemostatic system, blood pressure, muscle tissue, and extracellular matrix integrity [15,16,17,18]. Among them, *B. jararaca* venom is especially hypotensive [19,20]. Specific envenoming manifestations caused by viperid venoms include local swelling, blistering, hemorrhage, spontaneous bleeding, and in severe cases, necrosis and acute renal failure [18,21,22,23]. Furthermore, though rare, there are chronic consequences for viperid envenoming in humans, known as late effects or long-term sequelae, which are as yet poorly understood. They are quite variable and can occur days, months, or even years after the snakebite. Those effects include the reopening or persistence of wounds and random bleeding [24,25,26,27,28]. The mechanisms behind these phenomena may involve factors additional to the action of toxins. A few studies reported the presence of extracellular vesicles (EVs) in the snake venoms [29,30,31,32,33]. EVs, in general, have been described as important for cross-communication between organisms [34,35,36,37,38]. The terms cross-organism and cross-kingdom communication were conceived to refer to the interspecific interactions separate from the classic ecological relations, describing the communication at the molecular and cellular levels [39,40]. Thus, we hypothesize that snake venom EVs may participate in the spread and long-term effects of the toxins in the envenomed organism.

EVs are membrane vesicles carrying specific cargos, such as proteins, nucleic acids, and lipids, which modulate recipient cell function and biological processes [34,35,36,37,38]. Every living unicellular or multicellular organism secretes vesicles [41,42]. In multicellular organisms, different cell types and tissues secrete and generate heterogeneous populations of EVs that are accumulated in biological fluids, such as blood, urine, synovial fluids, milk, semen, etc. [35]. EVs can travel long distances within the same organism. They are crucial for homeostasis and pathological conditions (e.g., immune system communication and cancer metastasis). They are also involved in cross-organism communication through inter-species interactions by feeding [43,44], infection [39,45,46,47], and envenoming, as reported for wasp and spider venoms [48,49,50].

The outcome of EV modulation depends on the recipient cell and the EVs cellular origin [34,35,36,37,38]. Many pathogens vs. host recipient cells crosstalk result in the modulation of pathogen adhesion and infectivity, allowing the transference of toxic molecules and virulence factors, increasing pathogen survival and immune system escape [39,46,47,51,52]. For example, in Drosophila, EVs from the parasitoid wasp venom suppress immune cells, increasing the wasp’s offspring survival and parasitism success [48,53,54,55].

The knowledge of EVs found in snake venom is still limited. Using electron microscopy, Mackessy observed vesicles in the lumen of the venom gland from *Crotalus viridis oreganus* in a region close to the secretory epithelium [56]. Carneiro et al. made the same observations in the *Crotalus durissus terrificus* venom gland and characterized them as microvesicles for the first time [29]. EVs were also found in *Gloydius blomhoffii blomhoffii* venom [30]. This last study was driven by previous identification of toxins with transmembrane domains, suggesting the occurrence of unconventional protein secretion in *Gloydius* venoms, possibly related to EV secretion [57,58].

Souza-Imberg et al. described the EV proteome of *Crotalus d. terrificus* venom using in-gel digestion of proteins [31] and Carregari et al. performed the proteomic analysis of EVs from four viperid species: *Agkistrodon contortrix contortrix*, *Crotalus atrox*, *Crotalus viridis,* and *Crotalus cerberus oreganus* [32]. Although high-throughput proteomic was firstly applied to EV analysis, this study ended up identifying mainly the classic venom toxins, other than EV proteins [32]. Willard et al. complemented the protein list identified from *Crotalus atrox* and *Crotalus oreganus helleri* EVs, including plasma biomarkers from envenomed mice [33]. Taken together, these studies gave insights into snake venom EV biology, but their biological relevance is yet to be defined. Thus, studying the composition and effects of *B. jararaca* EVs could impact the current understanding of envenoming and snake vs. prey interactions.

In the present work, *B. jararaca* EVs (Bj-EVs) were purified from fresh crude venom. Several conserved EV markers and many types of processing enzymes were identified by proteomic analysis. Using fluorescence and electron microscopy, we demonstrated that mammalian cells uptake Bj-EVs, which establishes a basis for discovering new mechanisms of interaction between venoms and recipient cells, further indicating the potential of cross-organism communication. Based on these findings, we hypothesize that venom-EVs may play a role in toxin processing and stability in the venom gland.

## 2. Results

### 2.1. Bothrops Jararaca Venom Contains Distinct Extracellular Vesicles Populations

In the present work, Bj-EVs were isolated from pools of fresh venom by classic differential centrifugation (Appendix A) [59,60]. Briefly, the crude venom was submitted to sequential centrifugation at 20,000× *g* and 100,000× *g*, resulting in two pellets known as P20K and P100K, respectively. The ultrathin sections of P20K and P100K were visualized by transmission electron microscopy (TEM), revealing distinct vesicle groups (Figure 1A–H). It was possible to observe a conserved vesicular structure, including the lipid bilayer membrane (Figure 1D,H). P20K was a markedly diverse population, with particle sizes ranging from 30 nm to over 1,000 nm. Some structures presented intense electrondense granules, while others exhibited electronlucent content (Figure 1A–D). Despite the occurrence of large vesicles, the vast majority were sized ~100 nm. In contrast, P100K was a highly homogeneous population in terms of morphology, ranging from 30 to 130 nm.

The size of Bj-EVs was further measured using negative staining microscopy images individually (Figure 1I and Appendix A) and nanoparticle tracking analysis (NTA) data (Figure 1J). Although P20K and P100K populations showed a similar size distribution with peaks of ~100 nm, P20K has a higher proportion of >100 nm particles than P100K (Figure 1I,J). Interestingly, vesicles with similar sizes in each population differed in their staining properties (Appendix A).

### 2.2. Bj-EVs Exhibit Diverse Protein Composition, Conserved EV Biomarkers, and Their Venom-Related Proteins Signature

Bj-EV proteins represent a minor fraction (<0.1%) of the crude venom. The P20K fraction is at least five-fold more abundant in proteins than P100K. In parallel, NTA particle counting indicated approximately two-fold more particles in P20K than in P100K (Figure 1J), indicating that P20K has a higher number of proteins per particle. In fact, P20K particles had higher densities and were sedimented at a lower centrifugation speed than P100K.

Different protein profiles were observed in the SDS-PAGE (Figure 2) when comparing P20K and P100K fractions with the fresh crude venom and the venom depleted of vesicles (SP100K, the supernatant obtained after collecting P20K and P100K EVs, Appendix A). The P20K and P100K electrophoretic profiles showed high molecular weight proteins. The most intense electrophoretic band, with ~70 kDa of molecular weight, was observed in both P20K and P100K fractions (Figure 2, asterisks). The mass spectrometry analysis of the digested proteins in the ~70 kDa bands showed 5′-nucleotidase (5NTD) as the major protein present in Bj-EV fractions (Appendix A).

A proteomic analysis was conducted using a high-resolution mass spectrometer and a customized database, which included toxic and non-toxic protein sequences from several snakes, including *B. jararaca*. In total, we confidently identified 506 and 435 proteins in the P20K and P100K fractions, respectively. The crude venom (CV) and P100K supernatant (SP100K) resulted in 280 and 277 identified proteins, respectively. The variety of protein families was greater in P20K (185) and P100K (154) compared to the CV (18) (Figure 3A,B and Appendix A). As mentioned, *B. jararaca* venom is composed of principal toxin families such as SVMP, CTL, SVSP, and PLA_2_. Thus, the number of protein identifications obtained in the CV assigned for the specific subset of protein families is in accordance with other reports [10,11,12,13,14] (Figure 3A; Appendix A). Since Bj-EV fractions represent a small part of the CV, as expected, the venom depleted of vesicles (SP100K) resulted in the identification of 277 proteins, which almost overlapped with those identified in the crude venom (Figure 3 and Appendix A). Together, these results expand the molecular diversity knowledge for *B. jararaca* venom.

A principal component analysis (PCA) of proteomic data revealed that P20K, P100K, CV, and SP100K are distinct groups. We observed that CV and SP100K almost overlapped, while P20K and P100K were far apart from each other and from the CV-SP100K cluster (Figure 3C). Comparing P20K and P100K fractions, 130 and 59 proteins were exclusively found in each sample, respectively (Appendix A). Furthermore, there were significant quantitative differences in the 66.5% of P20K, and P100K shared identifications, as a zinc transporter zip4-like protein accumulated in P20K and a dnaj homolog subfamily c member three precursor accumulated in P100K (Figure 3D; Appendix A). All qualitative and quantitative information gathered from P20K and P100K reinforces that they are two distinct groups.

Although *B. jararaca* CV and its fractions (P20K, P100K, and SP100K) shared some toxin families, they differed in protein abundances (Table 1; Appendix A). For example, the major venom components, including SVMPs, CTLs, SVSPs, and LAO, are more abundant in the CV, while the phosphodiesterase (PDE) and 5NDT toxins are up-accumulated in Bj-EV fractions (Table 1). Furthermore, the 5NTD family constituted 19% and 13% of the total proteins in P20K and P100K, respectively (Appendix A).

Bj-EVs showed several well-described EV markers [61,62,63], including cytoskeletal proteins, HSPs, chloride intracellular channel protein 1, syntenin, annexins, and Ras-related proteins, confirming P20K and P100K EV origin. Additionally, Bj-EVs have venom gland-related proteins, such as toxins, PLA_2_ inhibitors, and peptidylprolyl isomerases (Table 2; Appendix A). Interestingly, Bj-EVs showed RNA-binding proteins (Appendix A).

The top enriched families in Bj-EVs included aminopeptidases, 5NTD, and phosphodiesterase (Appendix A and Table 2), which have also been found in other snake venom EVs and occurred in low abundance in other whole snake venom proteomes [30,31,32,64]. Those proteins are not general EV markers (Appendix A). Thus, in this work, we propose this set of proteins, which includes proteins other than the top enriched families, as venom-related EV markers (Table 2).

A dipeptidase-2 (DPEP-2) homologous protein was identified in the *B. jararaca* crude venom and all P20K, P100K, and SP100K fractions. It showed a significant increase of ~5.0- and ~1.9-fold change in protein amounts at P20K and P100K compared with CV, respectively (Table 1; Appendix A). DPEP-2 has been identified in a few venom proteomes [64,65]. The presence of DPEP-2 and other *B. jararaca* EV proteins, such as aminopeptidase A (APA), aminopeptidase P, and DPP-IV, indicate a pattern of processing enzymes enriched in snake venom EVs (Table 2).

### 2.3. SvEVs Interact with Mammalian Cells

To investigate the possible roles of Bj-EVs in cross-organism communication, the interaction between the P20K fraction and mammalian cells was analyzed by fluorescence and electron microscopy. Macrophages (Figure 4) and muscle cell lines (Figure 5) were exposed to Dil-labeled P20K. Both in macrophages and muscle cell lines, the P20K vesicles were internalized in a time-dependent manner (Figure 4 and Figure 5). The internalization signal could be observed only after the cells were treated with Dil-labeled EVs for longer than 30 min.

After four hours of P20K interaction with both macrophages and muscle cells, it was possible to observe a small number of vesicles inside the cells (Figure 4B and Figure 5B, arrows). After 24 h of exposure, we observed a significant increase in the P20K signal (Figure 4E and Figure 5E, arrows), indicating EV internalization. The macrophages exposed to P20K showed changes in the cytosol and overall cell architecture compared to non-treated cells (Appendix A). Notably, after 24 h of exposure, they became more vacuolized, and the fluorescence signal of P20K seemed to be transferred to another cell membrane, indicating EV membrane fusion, processing, and/or recycling (Figure 4). Fluorescence images of muscle cells revealed the colocalization of EVs with actin filaments observed by the red and green overlapped regions, as indicated by the yellow color (Figure 5C,F).

Pearson’s correlation from the median plane of the 3D deconvolution data (Figure 6A–H) and the reconstructed cell volumes represented through axis projections (Appendix A) provided further evidence for the presence of P20K inside the cells. Additionally, the axis projections allowed the observation of vesicles accumulated close to the nuclei (Appendix A).

The colocalization rate (Figure 6A–H) showed that macrophages uptake more EVs than muscle cells, which agreed with the flow cytometry quantification data (Figure 6I). In the images obtained after Pearson’s correlation analysis (Figure 6B,D,F,H), the signals for EVs are represented in red, for actin in green, and the overlap areas in yellow. For both cell lines tested, the intensity of the green signal increased after 24 h of experiment (Figure 6B,D,F,H), which may indicate a remodeling of actin filaments. The red and yellow signal intensities for macrophages increased after 24 h, indicating that the vesicles were incorporated into structures that colocalize with actin filaments. The same colocalization was observed for the muscle cells, although the red signals were lower in intensity.

To quantify the cellular uptake of P20K EVs, muscle and macrophage cells were exposed for 24 h with Dil-labeled EVs, and the end-point fluorescence intensity was measured by flow cytometry. Consistent with the result that macrophages uptake more vesicles than muscle cells (Figure 6F,H), approximately 61% of the macrophages and 20% of the muscle cells were positive for Dil fluorescence signal after 24 h (Figure 6I). Interestingly, the pre-incubation of P20K with the antibothropic serum (ABS, commercial antivenom) inhibited vesicle uptake in both cell lines (Figure 6I). However, not all P20K proteins were recognized by the antivenom (Appendix A). Since the ABS was incubated with intact vesicles, in which most surface proteins were exposed, this result indicates that EV surface proteins are required for vesicle internalization. In addition, the ABS was more efficient in blocking the EV internalization by muscle cells than macrophages, with relative inhibitions of 72% and 49% of EV uptake (Dil-fluorescence positive events), respectively (Figure 6I).

To identify other morphologic changes in muscle and macrophage cell lines after Bj-EV uptake, cells treated with P20K and P100K were analyzed using electron microscopy. In both cases, several EV-uptake events, including phagocytosis and micropinocytosis and numerous vesicles localized close to the plasma membrane, were observed, mainly after four hours of treatment (Figure 7, Figure 8 and Appendix A, arrowheads). In agreement with other reports in which EVs localize in endo-lysosomal compartments [66,67,68], many treated cells showed endosome compartments filled with vesicle-like structures (Figure 7B, Figure 8B and Appendix A, arrows).

Ultrastructural changes were more evident in the cytosol and nucleus of muscle and macrophage cells after 24 h of EV exposure. It was possible to observe changes in the nuclear envelope shape and a reduction in chromatin compaction (Figure 7B,D and Figure 8B,D), more frequently found for P100K treated cells, while the control cells clearly showed typical heterochromatin (more dense staining) and euchromatin (Appendix A). After four and 24 h of treatment, both cell lines showed an increased number of cytosolic vacuoles and mitochondrial morphology changes (Figure 7, Figure 8, Appendix A and Appendix A).

## 3. Discussion

Despite the extensive studies of *B. jararaca* snake venom in recent decades, which revealed several intraspecific differences in terms of composition and biological activities [13,14], this is the first report of extracellular vesicles in *B. jararaca* snake venom.

Previous works using size exclusion chromatography, one-step ultracentrifugation, or magnetic beads for EV isolation reported a single fraction of EVs from other snake venoms [29,30,31,32,33]. In this study, two EV populations were isolated from *B. jararaca* venom by differential ultracentrifugation (Appendix A). Together, morphologic (Figure 1) and proteomic data (Figure 2, Figure 3 and Appendix A) support the existence of two different EV populations, P20K and P100K. Such heterogeneity is expected in EVs isolated from biological fluids [35]. Many EV types have been discovered recently, such as ARMMS, migrasomes, and mitovesicles [69,70,71], as well as extracellular nanoparticles (which are membranelles), such as exomeres and supermeres [72,73]. Among them, microvesicles and exosomes are well described in the literature and classified according to their cellular origin. Microvesicles are released directly from the plasma membrane and have a size range of 100–10,000 nm. Multivesicular bodies (MVBs) are members of the endosomal system and have many intraluminal vesicles (ILVs) ranging from ~30–150 nm. When MVBs fuse with the plasma membrane, exosomes are released into the extracellular space [34,38].

Souza-Imberg et al. observed small vesicles in the lumen of the venom gland of *Crotalus durissus terrificus*, budding directly from the plasma membrane of secretory cells and in the collected fresh venom after one-step centrifugation [31]. Although exosomes could not be excluded from the EV pool, the observation of budding vesicles in the *C. d. terrificus* venom gland strongly indicates the presence of microvesicles in this venom. In the case of Bj-EVs, the biogenesis was hypothesized based on the identification of some EV markers in these fractions (Appendix A). EV markers are recurrently identified proteins enriched in certain types of EVs compared with their parental cells. The EV markers can be either membrane or cytosolic proteins that are specifically loaded in EVs during their biogenesis [34].

Among the EV markers identified in Bj-EVs, syntenin-1 is a previously demonstrated exosome or small EV biomarker [74,75,76], which plays an important role in EV cargo sorting and membrane budding by interacting with syndecans and ALIX [76,77]. The chromatin modifier protein 5 (CHMP5)/Vsp60 is responsible for membrane scission events together with Vsp4 and ALIX ESCRT-III-associated proteins. Syntenin-1 and ALIX were identified in both Bj-EV fractions, while CHMP5 was detected only in P100K. In parallel, the microvesicle-enriched protein Annexin A1 was identified in the P20K fraction [38,78]. The proteomic identification of EV markers and the morphological characterization presented here indicate that P100K is likely a small EV-enriched fraction, while P20K is a more heterogeneous EV population, composed, at least in part, by microvesicles.

In the present work, we classified some EV-enriched proteins found in *B. jararaca* and other snake venoms as venom-related EV markers, such as 5′-nucleotidase (5NTD), dipeptidyl-peptidase IV, phosphodiesterase, and aminopeptidases (Table 2). These proteins were already known as low abundant toxins in snake venoms and were previously described in crude venom proteomes and venom gland transcriptomes [79,80,81,82]. In addition, most of them have a membrane-protein topology [58,83] and were reported in other snake venom EVs [30,31,33]. The biological function of these toxins is controversial. For instance, a few 5NTDs and PDEs were isolated from snake venoms and showed blood clotting and platelet aggregation inhibitory activity [84,85,86,87,88,89]. Hypotheses supporting their hypotensive effects were also published [90,91,92,93]. On the other hand, orthologous proteins of the dipeptidases, aminopeptidases, dipeptidyl peptidase, and serine carboxypeptidase (Table 2) have a consolidated biological role as processing enzymes [94,95,96,97].

Based on the identified venom-processing EV markers, we hypothesize that Bj-EVs may be involved in processing venom components and/or processing prey proteins. Ogawa et al. isolated EVs from *Gloydius b. blomhoffi* snake venom and demonstrated that those EVs had processing activity on physiological oligopeptides, i.e., angiotensin II, glucose-dependent insulinotropic peptide, and glucagon-like peptide-1 [30]. These results indicated the possible involvement of EVs in disturbing physiological processes after envenomation, such as blood pressure and glucose homeostasis imbalance [30]. Additionally, snake venom EVs may be involved in toxin processing and maturation inside the venom gland.

In viperid snakes, venom is stored for long periods in the extracellular space, in the large venom gland basal lumen [98,99]. Several toxins lose their prodomain and undergo maturation and post-translational modifications after reaching the lumen of the venom gland. Typically, SVSPs and SVMPs are secreted as zymogens [100,101,102]. The bradykinin-potentiating and C-type natriuretic peptides (BPP-CNP) are toxins expressed as a large precursor cleaved in many mature peptides [103,104,105]. Other toxins, such as *B. jararaca* disintegrins, have proteoforms identified in the crude venom in which the N or C-terminus amino acids are missing, with the unprocessed and processed forms coexisting in the toxin pool [14,106,107]. Portes-Junior et al. observed that the toxin jararhagin, an SVMP from *B. jararaca* venom, is secreted as a zymogen. Its prodomain is significantly cleaved only when the toxin reaches the lumen of the venom gland [108]. In another study with primary cultures of secretory cells, SVMPs were identified in their zymogen form in the culture medium, in opposition to what is observed in natural venom. These findings indicated that the processing molecules are not supplied by the same toxin-producing cells, supporting the assumption that processing occurs after protein secretion in the lumen of the venom gland [109]. However, the underlying mechanisms and the enzymes involved are not clear. Thus, based on the proteins found enriched in Bj-EVs (Table 2), i.e., aminopeptidases, dipeptidases, and carboxypeptidases, we propose that Bj-EV proteins may be involved in toxin processing (post-translational modification, zymogen cleaving, and toxin maturation) and overall venom stability (presence of isomerases and PLA_2_ inhibitor).

The 5NTD is the most abundant protein family in Bj-EVs. Many 5NTDs from snake venoms are homologous to the ecto-5′-nucleotidases, also known as CD73, which carry a GPI-anchor that attaches proteins to membranes [88,110,111,112]. The first report of EVs in the literature described high 5′-nucleotidase activity, and the authors hypothesized a role in the dephosphorylation of membrane constituents of recipient cells [113]. Since then, CD73 has been described as enriched in many other EVs, for example, the EVs from immune and tumor cells that play a role in immune suppression [114,115,116,117].

In many types of cells from diverse organisms, CD73 works with other ecto-nucleotidases, including CD39 (NTPdases) and PDEs (NPPs), in extracellular purinergic signaling. CD39 and PDEs release AMP from ATP and ADP; subsequently, CD73 releases free adenosine (Ado) from 5′-AMP [115,116]. In the immune system of mammals, Ado acts through adenosine receptors, inducing immune suppression by blocking the activation and proliferation of lymphocytes and neutrophils while impairing cytokine secretion in macrophages [117,118,119,120]. Furthermore, the super expression of CD39 and CD73 in cancer cells promotes immune escape, tumor survival, and metastasis [114,116,121,122,123,124]. In this context, these nucleotidases have been considered pharmacological targets for cancer treatment [112,115,120,125].

In the case of snake venoms, 5NTD may be acting synergistically with another enriched Bj-EV protein, the PDE, generating Ado in the venom and/or in the blood circulation of the envenomed organism. Some authors proposed the existence of purinergic hypotension caused by snake venoms [90,91,92,93]. In this perspective, Ado would act as an acute hypotensive molecule. However, these mechanisms were not demonstrated experimentally or with the involvement of EVs.

The Ado pharmacological effects differ based on the purinergic receptors type and in which cell lineage they occur. Ado can cause vasodilation via endothelial A_2_A and A_2_B receptors, increase vascular permeability by mast cell activation (A_3_), provoke central sedative effects via neuronal A_1_ receptors, and bradycardia through A_1_ receptor, which together can contribute to hypotension and paralysis [90,126,127]. Considering the aspect of vascular permeability, Bj-EVs carrying 5NTD could contribute to venom spreading, since soluble 5NTD from snake venom causes vascular leakage in vivo [128]. On the other hand, Ado is immunosuppressive when immune cells express P1 receptors [117,118,119,120,129], which may be related to the long-term effects of snakebites [24,25,26,27,28]. In accordance with this, a strong immune suppressive function is well characterized in EVs from wasp venoms [48,54,55].

5NTD is ubiquitous in snake venoms [79,91], is a minor toxin fraction observed in venom proteomes [14,130,131], is enriched in snake venom EVs (as demonstrated here and in previous works [30,31,33]), and is usually predicted to have a GPI-anchor. Based on this knowledge and in our results, we suggest that a fraction of 5NTD is structurally related to EVs. Furthermore, we speculate that EVs may be part of all snake venoms. The presence of free nucleosides and nucleotides in high abundance in snake venoms (which can exceed 8% of their dry weight) corroborates these claims [3,90,91,132]. EVs have previously been isolated from venoms of other taxons, such as wasp and spider venoms, indicating their presence in this type of biological fluid.

Lipophilic dyes have been extensively used for EV labeling for in vitro and in vivo uptake assays [133,134,135]. In this study, fluorescence microscopy and flow cytometry revealed that P20K vesicles are internalized by muscle cells and macrophages. Interestingly, the pre-incubation of P20K with ABS reduced the cellular uptake of EVs (Figure 6I). The ABS is a polyclonal serum obtained after immunizing horses with a mixture of crude venoms collected from five *Bothrops* snakes: *B. jararaca* (50%), *B. jararacussu* (12.5%), *B. moojeni* (12.5%), *B. alternatus* (12.5%), and *B. neuwiedi* (12.5%). ABS was shown to react and neutralize the main *Bothrops* snake venom toxins [14,118,119]. Our results suggest that specific EV surface proteins are important for internalization and that the ABS may partially protect the cells from EVs. As expected, Western blotting showed that the ABS recognized the entire CV protein profile. In contrast, Bj-EV proteins were partially recognized (Appendix A), which may be associated with the low abundance of Bj-EVs in the venom. In addition, the ABS recognized differentially the Bj-EVs obtained from distinct pools of animals (Appendix A). Similarly to what is known for crude snake venoms, which show individual intraspecific variability, especially in *B. jararaca* [14,136,137,138], these results indicate that the protein composition of Bj-EVs varies with each animal/venom sampling.

Even though macrophages were expected to uptake more vesicles due to their phagocytosis function, the ABS treatment blocked P20K uptake by muscle cells more efficiently than macrophages, inhibiting 72% and 49% of the EV internalization, respectively. This result indicates that distinct EV surface proteins could be required for EV uptake by different cell lines.

In electron microscopy, after 24 h of cell treatment with both P20K and P100K, cytosol vacuolization and disorganization, possible mitochondrial alterations, and abnormal chromatin condensation were observed. In addition, using fluorescence microscopy, we observed a change in the actin filament distribution, mainly in macrophages. Similarly, HUVEC cells treated with spider venom EVs also exhibited actin disorganization [50]. Whether these alterations contribute to the development of envenoming clinical effects is still an open question.

Macrophages are cells with high phagocytic ability. In most cases, these cells uptake circulating vesicles when administered in vivo [134,139,140]. The parasitoid wasp venom suppresses the *Drosophila* immune system, facilitating wasp offspring development and survival [48,53,141,142]. Notably, the macrophage-like immune cells of *Drosophila* internalize the wasp venom EVs that circulate in the hemolymph, which leads to the destruction of the phagolysosomal compartments and cause their death by apoptosis [54,55]. Based on the CD73^+^PDE^+^ phenotype found in Bj-EVs, we expect that the vesicles can modulate macrophages and other immune cells in vivo. The same may happen for muscle cells, as the cellular alterations observed in the A7R5 aortic smooth muscle cell line may also represent a possible participation of this tissue in the pathophysiology of the envenoming. Viperid venoms affect the endothelial cells and the integrity of blood vessels causing edema, hemorrhage, and extravasation of plasma to the extravascular space [16,17,143,144]. These effects are due to direct cellular damage, increased vascular permeability, and basement membrane degradation. Alongside this, myotoxicity and myonecrosis are relevant aspects of a viperid snakebite [17,22]. Accordingly, the skeletal muscle tissue may also be a target of Bj-EVs. These results, together with the insights regarding the interaction of Bj-EVs with mammalian targets in vivo, will certainly provide a more complete picture of animal response to envenoming.

Snakebite envenoming poses a threat to life as it can cause an acute inflammatory response, ranging from debilitation to death. The possible effects of snake venom EVs on the prey/victim immune system and other tissues may be linked to the less frequent cases of late effects and chronic manifestations. The enrichment of RNA-binding proteins in Bj-EVs, such as Y-box-binding protein 1, polyadenylate-binding protein, regulator of nonsense transcripts 1, and the heterogeneous nuclear ribonucleoprotein Q (Appendix A), supports the assumption that they also contain nucleic acids [145,146]. In fact, mRNAs have been unexpectedly found in snake venoms [147,148,149]. The occurrence of mRNA in EVs could explain how such labile molecules remain undegraded in venoms. Since EV RNAs from other organisms have been shown to reprogram the recipient cell phenotype [150,151], we will investigate Bj-EV RNAs and how they may contribute to the understanding of envenoming.

We have proposed some varied possible biological functions related to Bj-EVs. They are all reasonable, as venoms have evolved to be multifunctional and are able to have several molecular targets at the same time [152]. Additionally, snake venom EVs may be important to protect their cargo from proteases, which requires further investigation. The present results lay a foundation for further studies on the biological effects of snake venom EVs.

## 4. Conclusions

In this study, we identified a heterogeneous EV population from *Bothrops jararaca* venom. The morphological analysis revealed that Bj-EVs are nanoparticles, mostly ranging from 50 to 300 nm, and have a real-vesicular structure with lipid bilayer membranes. We identified conserved EV markers alongside specific venom-related proteins that are enriched in the EV samples instead of the crude venom. Interestingly, the venom-related EV markers are composed of processing enzymes and proteins related to protein folding and stability. Based on these findings, we proposed that Bj-EVs may be participating in the maturation of toxins in the lumen of the venom gland.

In vitro studies showed that mammalian cells uptake Bj-EVs. Macrophages uptake more vesicles than muscle cells and the ABS partially inhibits EV internalization in both cell lines. In addition, the cells exposed to EVs showed ultrastructural changes, such as cytoskeleton reorganization, alterations in chromatin compaction, and mitochondrial morphology. Based on the CD73^+^PDE^+^ phenotype and the macrophage interaction with the vesicles, we hypothesized that Bj-EVs may participate in immune modulation in human envenoming. Further studies using animal models are necessary to reveal the biodistribution of EVs in vivo and to elucidate the biological role of snake venom EVs in envenoming and cross-organism communication. Nonetheless, our results emphasize the presence of vesicles in venoms, giving insights into their possible biological role as the potential action in cellular modulation and in the processing of venom or prey proteins.

## 5. Materials and Methods

### 5.1. Bothrops jararaca Venom

Adult specimens of *Bothrops jararaca* maintained at the serpentarium of Instituto Vital Brazil (Niterói, Rio de Janeiro, Brazil) were previously anesthetized by CO_2_ inhalation. Fresh venom was manually extracted and pooled. Each extraction group contained at least 11 specimens including males and females, all native to the southeastern region of Brazil. Fresh venom pools were kept at 4 °C until extracellular vesicle isolation.

### 5.2. Extracellular Vesicles Isolation

After being centrifuged at 8,000× *g* for 25 min for cell debris removal, the venom was diluted in sterile 0.1 mM citrate buffer pH 5.0 containing 100 U/mL penicillin and 100 µg/mL streptomycin and was further centrifuged at 20,000× *g* for 25 min. The first EV pellet was collected (P20K) and the supernatant was centrifuged at 100,000× *g* for 2 h. The second EV pellet (P100K) was obtained and the P100K supernatant (SP100K) corresponded to the venom depleted of vesicles. The P20K and P100K samples were then washed in citrate buffer using the respective centrifugation speeds. The samples were considered clean from extravesicular proteins when the buffer absorbances at 280 nm reached near 0 values. After washing, EVs were resuspended in a new buffer, immediately frozen in nitrogen, and stored at −20 °C until use. The protein contents of the crude venom and venom fractions were quantified using a Qubit protein assay kit (Thermo Fisher Scientific, Waltham, MA USA ) (Appendix A).

### 5.3. Transmission Electron Microscopy (TEM)

P20K and P100K were washed twice with PBS pH 7.2 and fixed in 2.5% glutaraldehyde, 0.1 M sodium cacodylate buffer pH 7.2 for at least 1 h. After fixation, vesicles were washed in cacodylate buffer pH 7.2 and postfixed in a solution containing 1% osmium tetroxide, 1.25% potassium ferrocyanide, and 5 mM calcium chloride in 0.1 M cacodylate buffer for 30 min at room temperature in the dark. Next, pellets were washed in the same buffer, dehydrated in increasing acetone concentrations, and embedded in Epoxy resin. The first steps of fixation, post-fixation, and washes of P100K were conducted in a micro ultracentrifuge (Optima MAX-XP, TLA 55 rotor, Beckman Coulter) at 100,000× *g* due to the low density of the vesicle population. Next, ultrathin sections were obtained in an ultramicrotome (RMC Boeckeler, Tucson, AZ, USA), stained with 5% uranyl acetate and lead citrate solution.

For negative staining, isolated vesicles were directly resuspended in the fixation solution (2.5% glutaraldehyde in 0.1 M sodium cacodylate buffer pH 7.2), absorbed for 2 min in formvar/carbon-coated copper grids, and stained in 5% uranyl acetate for 30 s. Both samples were visualized in a FEI Tecnai Spirit microscope at 120 kV.

### 5.4. EV Size Analysis

The size of venom vesicles was determined by nanoparticle tracking analysis using ZetaView^®^ (Particle Metrix, Meerbusch, Germany). Samples were diluted in citrate buffer and analyzed in size distribution mode with 85% sensitivity and 55 shutter pre-acquisition. Post-acquisition parameters were set to a minimum brightness of 50 and a minimum and maximum size of 5 nm and 500 nm, respectively, using ZetaView 8.02.31 software. Exported data were normalized for initial particle concentration and plotted in GraphPad Prism 8.

Due to the wide size distribution with the remarkable dynamic range within vesicle populations, morphometric analysis was conducted in images of negative staining using Image J 1.50i (NIH, USA), dimensioning n = 705 and 174 particles in P20K and P100K, respectively.

### 5.5. SDS-PAGE and Western Blot

15 µg of *B. jararaca* venom fractions were loaded on a discontinuous system of 12% polyacrylamide gels under reducing conditions, stained with Coomassie Brilliant Blue G-250 (MP Biomedicals), and visualized in a ChemiDoc MP Image System (Bio Rad) in the EPI-fluorescent mode. For Western blotting, 10 µg from each fraction were submitted to the same electrophoretic conditions and transferred to 0.45 µm PVDF membrane (GE Healthcare) at 200 V for 2 h in a mini wet Trans-Blot system (Bio-Rad). Membranes were blocked with 5% BSA for at least 2 h. The first antibody was the polyclonal antibothropic antivenom (ABS) with dilutions of 1:4000 or 1:5000 from a 32 µg/µL solution for 2 h at room temperature (Instituto Vital Brazil, http://www.vitalbrazil.rj.gov.br, accessed on 10 September 2021). The second antibody was the anti-Horse IgG—Peroxidase (Sigma, SAB3700130) incubated for 1 h with a 1:5000 dilution. Membranes were revealed using Clarity Max ECL reagent (Bio-Rad) in the ChemiDoc MP Image System (Bio-Rad).

### 5.6. Proteomics

#### 5.6.1. In-Gel Protein Digestion and Mass Spectrometry

Protein bands of interest were sliced from a Coomassie-G250 stained SDS-PAGE gel and subjected to in-gel reduction (10 mM dithiothreitol), alkylation (55 mM iodacetamide), and overnight trypsin digestion at 37 °C (Promega). Tryptic digests were dried, redissolved in 3% acetonitrile and 0.1% formic acid solution, and submitted to LC-MS/MS using Q-TOF micro mass equipment (Waters Corporation, Milford, MA, USA).

Electrospray voltage was set at 3500 V, source temperature at 80 °C, and cone voltage at 30 V. Instrument control and data acquisition were conducted using a MassLynx data system (Version 4.1, Waters), and experiments were performed by scanning from a mass-to-charge ratio (*m*/*z*) of 400–2000 using a scan time of 1 s, applied during the whole chromatographic process. Data-dependent MS/MS acquisitions (DDA) were performed on precursors with charge states of 2, 3, or 4 over a range of 50–2000 *m*/*z* and under a 2 *m*/*z* window. A maximum of 3 ions were selected for MS/MS from a single MS survey obtained by collision-induced dissociation (CID). Exported MS data were analyzed in Mascot MS/MS Ions Search, with parameters described in Appendix A.

#### 5.6.2. Protein Digestion and Mass Spectrometry for Shotgun Proteomics

Protein samples (30 µg) were precipitated in cold acetone overnight at −20 °C. Then, samples were resuspended in 50 µL of 8 M urea, 2 M thiourea, and 2% sodium deoxycholate (DOC) in 25 mM ammonium bicarbonate pH 8.0. Incubation steps were followed at room temperature (RT) as described: protein reduction (10 mM DTT for 60 min), alkylation (55 mM iodoacetamide—IAA for 30 min), and IAA quenching (55 mM of DTT for 15 min). The last two steps were undertaken in the dark. Finally, urea was diluted to <1 M with 25 mM ammonium bicarbonate, and trypsin digestion (Sequencing grade, Promega) proceeded overnight at a ratio of 1:20 at 37 °C.

DOC was precipitated by solution acidification with 1% trifluoroacetic acid (TFA), repeating 2 centrifugations steps at 12,000× *g* for 30 min at 4 °C. Tryptic peptides were desalted in a homemade C18 column using POROS^®^ 50 R2 reversed-phase packing matrix (Applied Biosystems) and eluted peptides were dried under vacuum.

For LC-MS/MS analyses, samples were resuspended in 0.1% formic acid (FA), quantified by Qubit^®^ Protein Assay Kit (Thermo Fisher Scientific) and injected into a nano-HPLC system Easy-nLC 1000 (Thermo Scientific). First, at reverse-phase trap column (2 cm × 150 µm i.d., ReproSil-Pur C18 AQ, 5 µm, 120 Å, Dr. Maisch GmbH) followed by the analytical column (15 cm × 75 µm i.d. ReproSil-Pur C18 AQ, 3 µm, 120 Å, Dr. Maisch GmbH). The flow rate was set to 250 nL/min with 0.1% FA + 5% acetonitrile (solution A) and 0.1% FA + 95% acetonitrile (solution B). A linear gradient was developed with a flush time of 10 min with 100% A, followed by 5–40% B for 107 min, 40–95% B for 5 min, and isocratically 90% B for 8 min. Eluting peptides were analyzed online by electrospray-ionization tandem mass spectrometry (ESI-MS/MS) using a LTQ Velos Orbitrap (Thermo Fisher Scientific, San Jose, CA) controlled by Tune 2.7.0.1103 SP1 and Xcalibur 2.2 SP1.48.

A full scan was performed in high resolution (60,000) in the Orbitrap with 500 ms accumulation time and 350–2000 *m*/*z* range. In DDA mode, the 10 most abundant ions were selected for Higher Energy Collision Dissociation (HCD) with an isolation width of 2.5 *m*/*z*, normalized collision energy of 35, and 10 ms activation time. MS2 analyses were performed in the Orbitrap (7500 resolution) with 100 ms accumulation time and 45 s excluding time.

##### Data Analysis

-Peptide Spectrum Matching (PSM)

Data were analyzed with PatternLab for proteomics software v4.1.1.22, which uses the comet algorithm [153]. A database was created merging “Uniprot serpentes” downloaded on 22 March 2019, *B. jararaca* accessory gland proteome [154] and *B. jararaca* tissues 454-sequecing [155], resulting in 227,747 sequences. *B. jararaca* translated databases were kindly provided by the authors. Carbamidomethylation of cysteines was set as a fixed modification, while methionine oxidation was set as a variable modification. For the peptide search, the parameters were 3 trypsin missed cleavages, mass error tolerance of 40 ppm (for precursor ions), and 1.0005 bin (for fragment ions). PSM-generated data were processed and filtered using the Search Engine Processor (SEPro) [153,156]. The pre-processing quality filters included a Delta mass of 30 ppm and a Delta CN of 0.001. Proteins were assigned with a cutoff of 2 peptides and a minimum of 3 spectral counts, accepted based on a 1% false-discovery rate (FDR) at protein level. The final protein list was obtained by grouping proteins with maximum parsimony.

-Quantitative analysis

For a descriptive protein list and relative protein abundances in each sample, NSAF was used with a cutoff of 2 peptides and a minimum of 3 spectral counts. For relative percent abundances, contaminants were manually removed. Protein family clustering was also manually assembled (Appendix A). For quantitative comparisons between samples, XIC data were used to calculate the principal component analysis (PCA) (linear Kernel PCA) and to analyze differentially expressed proteins by the T-Fold algorithm [153,157]. Exported PCA coordinates, protein ratios, and quantitative *t*-test p-values were used to construct PCA and volcano plots in GraphPad Prism 8.

### 5.7. Immunofluorescence Microscopy

P20K vesicles were labeled using the lipophilic membrane marker Dil (Invitrogen^®^, #V22885) with 0.25 µL of Dil equivalent to 1 µg of total P20K proteins, for 30 min at 37 °C. Vesicles were washed 3 times at 20,000× *g*, for 10 min in Hepes buffer for excess dye removal. RAW 264.7 macrophages and A7R5 smooth muscle cell lines were seeded at a density of 5 × 10^4^ cells/mL in 24-well plates and cultivated for 24 h to the adherence. To synchronize the interactions and internalization of vesicles by cells, they were first maintained at 4 °C for 5 min. Then, labeled P20K (equivalent to 5 µg of total proteins) per well were added on ice, and plates were kept for 10 min at 4 °C, followed by incubation at 37 °C for 4 or 24 h.

After incubation with the labeled EVs, the cells were 3× washed with PHEM buffer pH 7.2 (25 mM MgCl_2_, 35 mM KCl, 5 mM EGTA, 10 mM HEPES, 30 mM PIPES). Then, they were fixed in a freshly prepared solution of 4% formaldehyde in 0.1 M PHEM for 4h at room temperature, washed 3 times, and subsequently permeabilized by incubation in acetone for 5 min at −20 °C. Next, cells were incubated in 50 mM ammonium chloride for 30 min, and then blocked in PBS buffer containing 3% BSA and 0.025% fish gelatin, pH 7.2, twice for 30 min or overnight at 4 °C. After blocking non-specific antigenic sites, cells were incubated with the Alexa Fluor 488^®^ phalloidin marker at a final concentration of 25 µg/mL for 1 h. Alexa Fluor 488^®^ phalloidin is a fluorescent marker conjugated to phalloidin with a high affinity for actin filaments. Finally, cells were incubated with Hoechst at 5 µg/mL for nucleus labeling, mounted on glass slides with n-propyl gallate, and sealed. Samples were observed using a Leica DMI 6000B fluorescence microscope, where optical cuts were made on the Z-axis followed by 3D deconvolution processing using LAS X software version v. 3.2.1.9702.

### 5.8. Bj-EV Uptake Quantification

To quantify the cellular uptake of vesicles, flow cytometry was performed for positive fluorescent events count. RAW 264.7 macrophages and A7R5 smooth muscle cell lines were seeded at a 5 × 10^4^ cells/mL density in 6-well plates. After 24 h of cultivation, cells were either treated with Dil-labeled P20K (equivalent to 5 µg of total proteins) or with vesicles preincubated with antibothropic serum (ABS) for 24 h. ABS in a 32 mg/mL solution was diluted at 1:1000, incubated with pre-stained P20K for 90 min at 37 °C, and then washed before cell treatment.

After incubation, cells were washed 3 times with serum-free medium and removed from the plates using a solution of 2% trypsin, washed in PBS pH 7.2 buffer, and resuspended in 500 µL of PBS pH 7.2 for flow cytometry analysis. In an Accuri C6 cytometer (Becton Dickinson, Franklin Lakes, NJ, USA), 15,000 events were evaluated, and the fluorescent population was measured compared to control cells without Dil-P20K treatment. Data were plotted and submitted to 2-way ANOVA statistical analysis using the GraphPad Prism 8.0.2 software (USA).

### 5.9. Transmission Electron Microscopy in Scanning Electron Microscopy (STEM-IN-SEM)

RAW 264.7 macrophages and A7R5 smooth muscle cell lines were cultivated in 25 cm^3^ culture flasks. Before cell confluence, they were treated with P20K or P100K (equivalent to 10 µg of total proteins) for 4 or 24 h. Control cultures received vesicle vehicles instead. Fixation, post-fixation, embedding, and ultramicrotomy were conducted as described above for TEM. Cells were visualized by transmission electron microscopy in scanning electron microscopy (STEM-IN-SEM) using a VEGA 3 LMU (Tescan Brno, Czech Republic) microscope, at 30 kV in STEM mode.

## Figures and Tables

**Figure 1 toxins-14-00806-f001:**
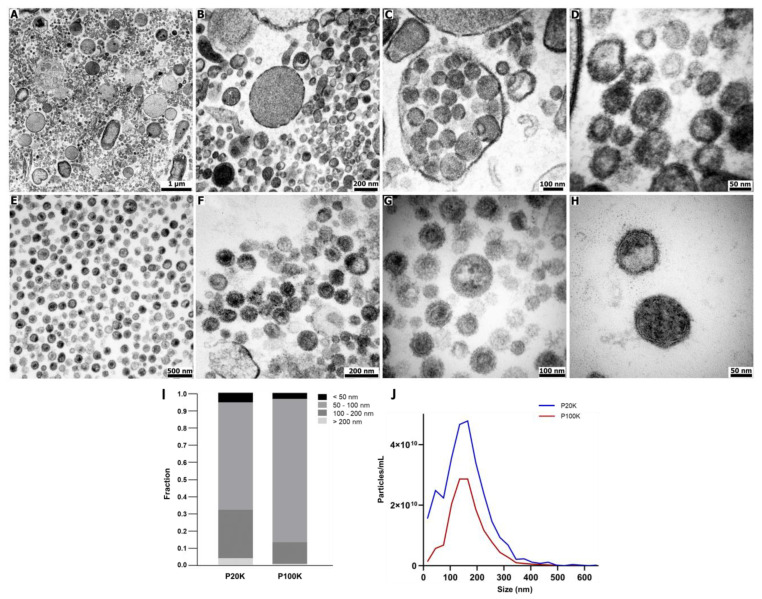
Morphological characterization of extracellular vesicles from Bj-EVs. Ultrathin sections of (**A**–**D**) P20K and (**E**–**H**) P100K were obtained by high voltage electron microscopy (Fei Tecnai Spirit, 120kV). (**I**) EV size distribution was determined by morphometric analysis of individual Bj-EVs. A total of 705 and 174 particles were analyzed from P20K and P100K, respectively, using negative staining images obtained by transmission electron microscopy. (**J**) Size distribution of Bj-EVs by nanoparticle tracking analysis (NTA) using a Zeta View (Particle Metrix). P20K = Bj-EVs pelleted at 20,000× *g*; P100K = Bj-EVs pelleted at 100,000× *g*.

**Figure 2 toxins-14-00806-f002:**
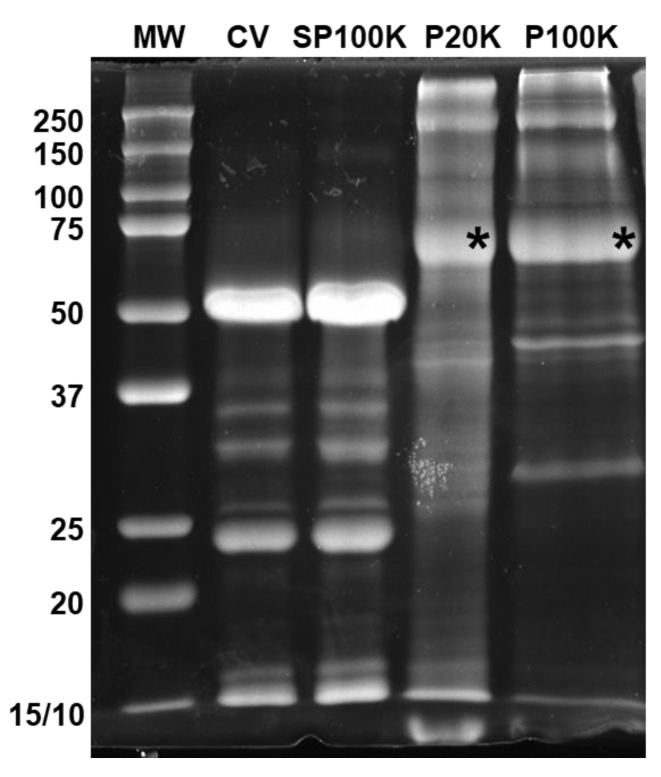
Protein profile of *B. jararaca* crude venom and its fractions. SDS-PAGE of *B. jararaca* crude venom and fractions (15 µg of protein) under reducing conditions, stained with Coomassie Brilliant Blue G-250. The asterisks highlight the protein bands digested and identified by mass spectrometry, described in Appendix A. CV = crude venom; SP100K = supernatant of P100K, or venom depleted of vesicles; P20K = vesicles pelleted at 20,000× *g*; P100K = vesicles pelleted at 100,000× *g*.

**Figure 3 toxins-14-00806-f003:**
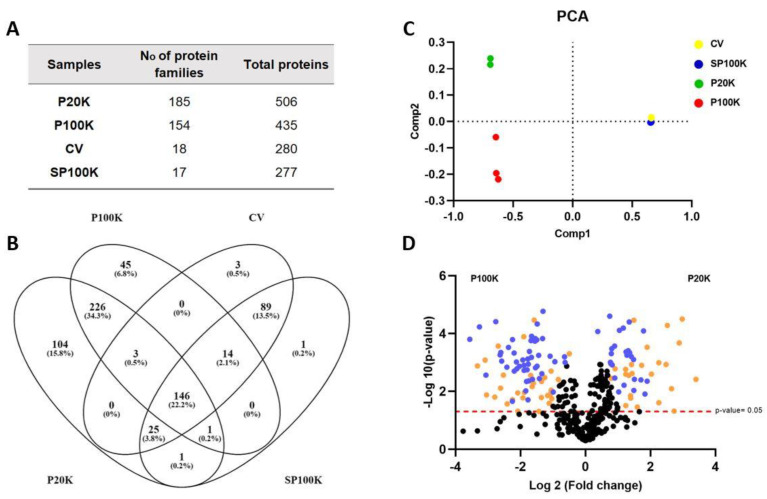
Shotgun Proteomics of *B. jararaca* crude venom and fractions. (**A**) Overview of the number of protein families and individual proteins identified in the samples. The full description of the identified protein families is listed in Appendix A. (**B**) Venn diagram showing common proteins among the crude venom and its fractions. (**C**) Principal component analysis (PCA) shows the global quantitative differences between the crude venom and its fractions. (**D**) Volcano plot showing the quantitative differences between P20K and P100K fractions. Blue dots represent highly significant quantitative differences between P20K and P100K. Orange dots are related to low abundance signals of proteins, filtered out by the L-stringency in the T-Fold node from PatternLab for Proteomics software, (q-value 0.05, F-stringency 0.10, L-stringency 0.40). CV = crude venom; SP100K = supernatant of P100K, or the venom depleted of vesicles; P20K = vesicles pelleted at 20,000× *g*; P100K = vesicles pelleted at 100,000× *g*.

**Figure 4 toxins-14-00806-f004:**
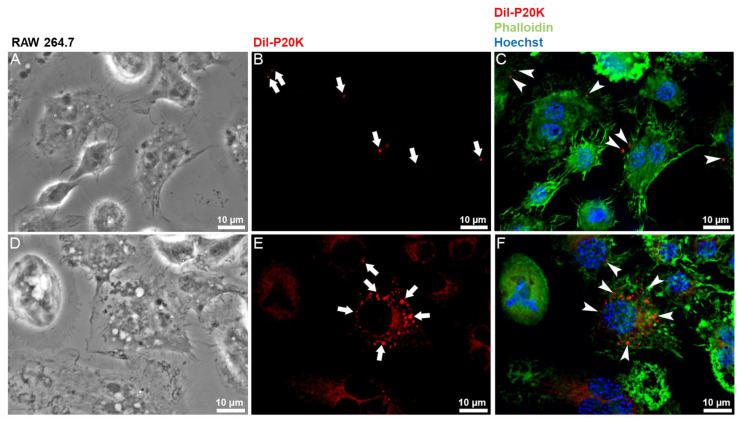
Fluorescence microscopy of RAW 264.7 macrophages treated with Dil-labeled vesicles. Macrophages were incubated with 5 µg of Dil-labeled P20K EV fraction for four hours (**A**–**C**) and 24 h (**D**–**F**). Images represent a maximum projection of images from different focal planes after the 3D deconvolution process. Cells maintained at the same condition but without P20K treatment (control) are shown in Appendix A. Red = P20K-EVs (Dil); green = actin (phalloidin); blue = nucleus (Hoechst). White arrows and arrowheads highlight internalized vesicles.

**Figure 5 toxins-14-00806-f005:**
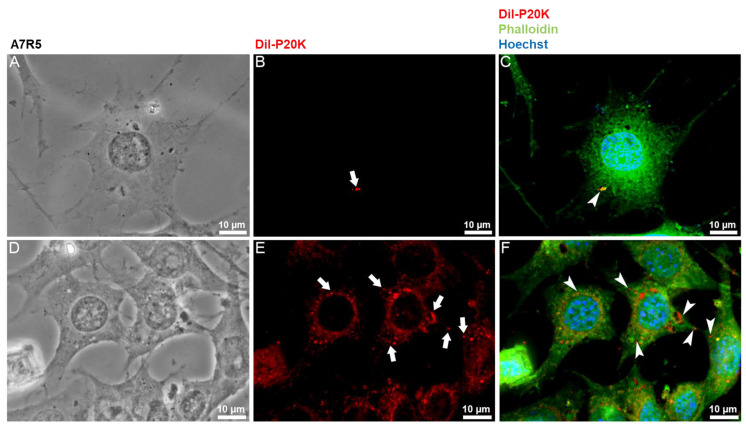
Fluorescence microscopy of A7R5 muscle cells treated with Dil-labeled vesicles. Muscle cells were incubated with 5 µg of Dil-labeled P20K EV fraction for four hours (**A**–**C**) and 24 h (**D**–**F**). Images represent a maximum projection of images from different focal planes after the 3D deconvolution process. Cells maintained under the same conditions but without P20K treatment are shown in Appendix A. Red = P20K-EVs (Dil); green = actin (phalloidin); blue = nucleus (Hoechst). White arrows and arrowheads highlight internalized vesicles.

**Figure 6 toxins-14-00806-f006:**
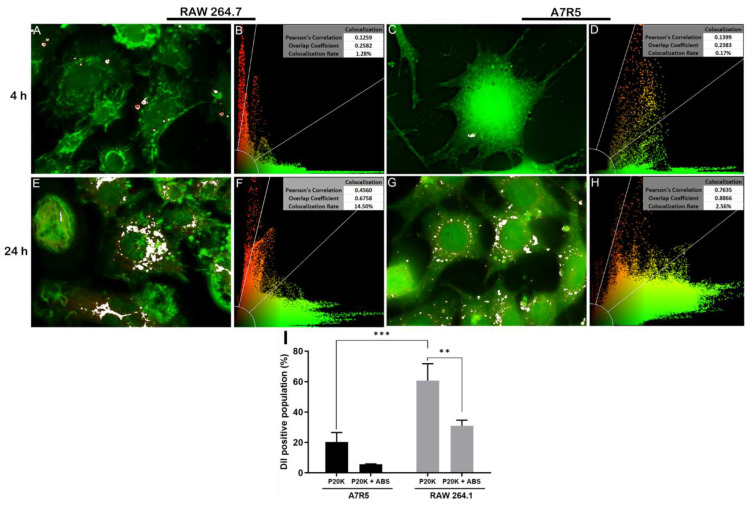
EV uptake in macrophages (RAW 264.7) and muscle cells (A7R5). (**A**–**H**) Pearson’s correlation analysis. White areas in A, C, E, and G and yellow areas in B, D, F, and H correspond to the overlap of P20K with the cytosolic region. (**I**) Fluorescence quantification of cells treated for 24 h with Dil-labeled P20K or Dil-labeled P20K pre-incubated with antibothropic antivenom (ABS). Statistical evaluation was performed by Tukey’s multiple comparisons test. *** *p* = 0.0003; ** *p* = 0.0025 red = P20K-EVs (Dil); green = actin (phalloidin).

**Figure 7 toxins-14-00806-f007:**
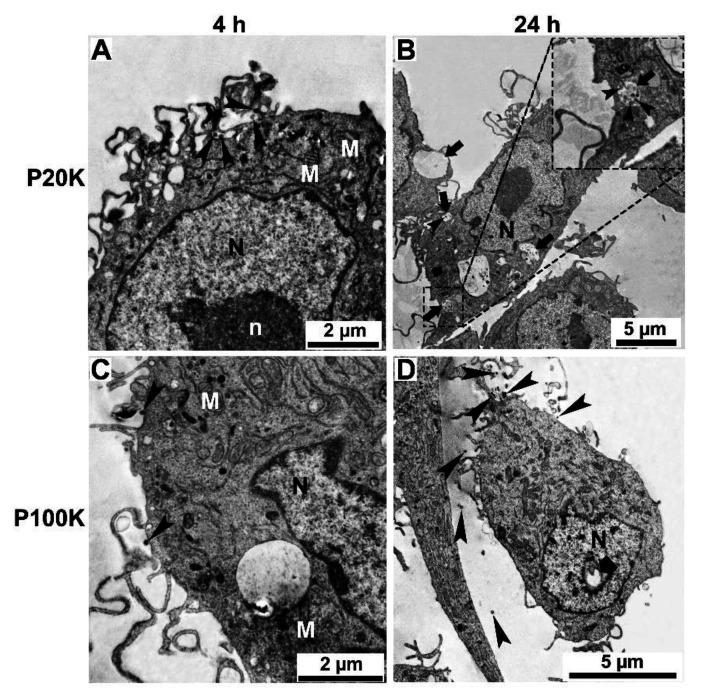
Ultrathin sections of RAW 264.7 macrophages treated with Bj-EVs. Macrophages were treated with P20K (**A**,**B**) and P100K (**C**,**D**). The arrowheads point to the EVs interacting with the cell surface or inside the cytosol. After four hours of interaction, several membrane projections can be observed (**A**,**C**). In addition, large vacuoles containing EVs (**B**, arrows) and alterations in the nucleus morphology and chromatin condensation (**B** and **D**, large arrow) were observed. Images were obtained by transmission electron microscopy in scanning electron microscopy (STEM-IN-SEM). N, nucleus; n, nucleolus; M, mitochondria.

**Figure 8 toxins-14-00806-f008:**
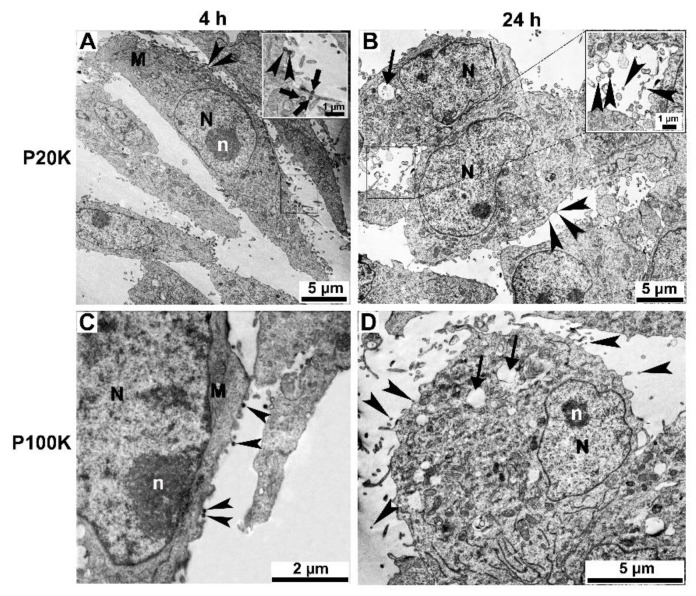
Ultrathin sections of A7R5 muscle cell line treated with Bj-EVs. Muscle cells were treated with P20K (**A**,**B**) and P100K (**C**,**D**). The arrowheads point to the EVs interacting with the cell surface. The inset on panel A shows a membrane projection. Arrows display large vacuoles (**D**) or vacuoles containing EVs (**B**). After 24 h of treatment, it is possible to observe abnormalities in the nuclear envelope morphology and chromatin condensation (**B**,**D**). Images were obtained by transmission electron microscopy in scanning electron microscopy (STEM-IN-SEM). N, nucleus; n, nucleolus; M, mitochondria.

**Table 1 toxins-14-00806-t001:** Enrichment of protein families in Bj-EVs relative to the *B. jararaca* crude venom. Overview of protein families identified or enriched in Bj-EVs and crude venom. X = protein not identified in the samples; ▲ = protein family enriched in the sample.

	Bj-EVs	Crude Venom
Major venom toxins	5NTD ▲	5NTD
	CTL	CTL ▲
	LAO	LAO ▲
	PDE ▲	PDE
	PLA_2_	PLA_2_ ▲
	SVMP	SVMP ▲
	SVSP	SVSP ▲
	X	svVEGF
Minor venom proteins	Aminopeptidase A	X
	Aminopeptidase P ▲	Aminopeptidase P
	Dipeptidase-2 ▲	Dipeptidase-2
	X	Glutaminyl-peptide cyclotransferase
	X	Hyaluronidase

**Table 2 toxins-14-00806-t002:** Snake venom-related EV markers. Protein families identified exclusively or enriched in Bj-EV fractions that are specifically related to venom. Most of these proteins were identified in other snake venom EVs.

Possible Function	Venom-Related EV Markers	Enriched Samples
Toxic or venom production	Ecto-5′-nucleotidase—5NTD	P20K, P100K
Toxic or venom production	Phosphodiesterase—PDE	P20K, P100K
Toxic or venom processing	Aminopeptidases (A, P)	P20K, P100K
Toxic or venom processing	Dipeptidases (DPEP-2, CNDP)	P20K, P100K
Toxic or venom processing	Dipeptidyl peptidase 4—DPP-IV	P20K, P100K
Toxic or venom processing	Dipeptidyl peptidase 3—DPP-3	P20K
Toxic or venom processing	Peptidase D	P20K
Toxic or venom processing	Serine carboxypeptidase CPVL	P20K
Venom maturation and stability	Peptidylprolyl isomerase	P20K, P100K
Venom maturation and stability	Disulfide-isomerase	P20K, P100K
Venom stability	PLA_2_ inhibitor	P20K

## Data Availability

The mass spectrometry proteomics data have been deposited in the ProteomeXchange Consortium via the PRIDE [156] partner repository with the dataset identifier PXD037375 and project name: characterization of extracellular vesicles from *Bothrops jararaca* snake venom.

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
