# Peer review of "Extracellular Vesicles from Bothrops jararaca Venom Are Diverse in Structure and Protein Composition and Interact with Mammalian Cells"

_toxins, 2022, doi:10.3390/toxins14110806_

Round 1
Reviewer 1 Report
In this work, authors studied extracellular vesicles isolated from B. jararaca venom.
Two fractions of EV have been obtained by centrifugation. Isolated EVs were analyzed by LC-MSMS, and proteins were identified by searching the customized snake proteins database. EV fractions showed different proteomes. Both EV fractions contained specific EV markers proteins and a subset called venom-related EV-markers (also present in other snake venoms EVs with low abundance in corresponding whole snake venom proteomes). Further, the effect of snake venom EVs on macrophages and muscle cell lines was investigated. The cytoskeleton, chromatin, and mitochondria were affected after the absorption of EVs by cells. The manuscript adds an interesting insight into the proteomics of snake venoms.
Minor points
Fraction P20k was used to quantify the cellular uptake of vesicles. Is there a particular reason, why only one EV fraction was selected for the examination of EV interaction with cells? Is it possible to expect a similar biological effect with the P100k fraction?
Labels (N, n, M) are difficult to recognize in Figures 7 and 8.
Author Response
Reviewer 1
In this work, authors studied extracellular vesicles isolated from B. jararaca venom.
Two fractions of EV have been obtained by centrifugation. Isolated EVs were analyzed by LC-MSMS, and proteins were identified by searching the customized snake proteins database. EV fractions showed different proteomes. Both EV fractions contained specific EV markers proteins and a subset called venom-related EV-markers (also present in other snake venoms EVs with low abundance in corresponding whole snake venom proteomes). Further, the effect of snake venom EVs on macrophages and muscle cell lines was investigated. The cytoskeleton, chromatin, and mitochondria were affected after the absorption of EVs by cells. The manuscript adds an interesting insight into the proteomics of snake venoms.
A: We thank the reviewer for the comments.
Minor points
- Fraction P20k was used to quantify the cellular uptake of vesicles. Is there a particular reason, why only one EV fraction was selected for the examination of EV interaction with cells? Is it possible to expect a similar biological effect with the P100k fraction?
A: Thanks for your observation. Indeed, we tried to label P100K in the amount demanded in the experiments. Nevertheless, due to the very low yield of this fraction (lines 137-138), we could not perform all the experiments, mainly related to the fluorescent probes that require many washing steps, lowing the yield even more.
- Labels (N, n, M) are difficult to recognize in Figures 7 and 8.
A: The figures were modified to label them adequately.
Reviewer 2 Report
The article “Extracellular vesicles from Bothrops jararaca venom are diverse in structure, protein composition and interact with mammalian cells” is a multidisciplinary study of the extracellular vesicles found in the venom of the snake Bothrops jararaca.
The authors observed these EVs by electron microscopy from venoms that were centrifuged at 20,000 and then 100,000 rpm. The analysis of the content of these vesicles shows that they contain proteins, in particular a molecule of about 70 kDa, proposed as a nucleotidase. Furthermore, different proteins are identified as being related to the venomous function. EVs interact with mammalian cells, notably macrophages and muscle cells: they are internalized in these target cells. This internalization could be inhibited by pre-incubation with an anti-venom specific to B. jararaca venom.
This study is interesting because it makes the link between the identification of Evs in a venom, and their possible function of maturation/transport of toxins to distribute them in a target organism. The experiments carried out show the heterogeneity of EVs, their content and the interaction they exert on muscle cells or marcophages.
Questions
L17: what do you mean by “known to be immunosuppressive in the context of EVs” ?
L59 “EVs have been described as important for cross-communication between organisms”: what do you call “cross-communication” in this context? The interaction between snake and prey at the molecular level? Make it clear.
Table S1: how can you explain the presence in EVs of proteins such as ion channels or HSPs, or signal transduction proteins which basically have a function in the cells? Are they really from EVs or could they just originate from cell debris? This point should be clearly addressed in the discussion.
Figs4-5: why not have performed the same experiment with raw venom to evaluate if Evs of the venom are not internalized as well, to place EVs in a more physiological situation?
Discussion: the authors propose that the function of EVs is linked to the processing of toxic proteins, i.e. their maturation to a final functional form, rather like hormones. However, the concept of toxicodynamic can be reinterpreted in the light of this work. It is indeed imagined that a venom contains protein toxins that are released during envenomation and transported as such in the internal environment of the target. This study suggests that toxins are transported to their physiological targets via EVs. This concept, in my opinion, needs to be further clarified to shed light on the functionality of EVs. Is the benefit of venom toxins being present in EVs to be protected from the target's resident proteases at the time of envenomation?
Author Response
Reviewer 2
The article “Extracellular vesicles from Bothrops jararaca venom are diverse in structure, protein composition and interact with mammalian cells” is a multidisciplinary study of the extracellular vesicles found in the venom of the snake Bothrops jararaca.
The authors observed these EVs by electron microscopy from venoms that were centrifuged at 20,000 and then 100,000 rpm. The analysis of the content of these vesicles shows that they contain proteins, in particular a molecule of about 70 kDa, proposed as a nucleotidase. Furthermore, different proteins are identified as being related to the venomous function. EVs interact with mammalian cells, notably macrophages and muscle cells: they are internalized in these target cells. This internalization could be inhibited by pre-incubation with an anti-venom specific to B. jararaca venom.
This study is interesting because it makes the link between the identification of Evs in a venom, and their possible function of maturation/transport of toxins to distribute them in a target organism. The experiments carried out show the heterogeneity of EVs, their content and the interaction they exert on muscle cells or marcophages.
A: We thank the reviewer for the comments.
Questions
- L17: what do you mean by “known to be immunosuppressive in the context of EVs”?
A: 5’-nucleotidase was demonstrated to be immunosuppressive in general and when attached to EVs [1]. For clarity, we changed the phrase in the abstract to a more general concept.
Line 18 “The most abundant protein family observed in the Bj-EVs was the 5’-nucleotidase, known to be immunosuppressive and a low abundant and ubiquitous toxin in snake venoms “
- L59 “EVs have been described as important for cross-communication between organisms”: what do you call “cross-communication” in this context? The interaction between snake and prey at the molecular level? Make it clear.
A: Thank you for this comment. We first wanted to introduce the general knowledge about EVs, and we confirm that this cross-communication occurs at the molecular level. To avoid any misunderstanding, we added the following phrase:
Lines 59-63 “EVs, in general, have been described as important for cross-communication between organisms. The terms cross-organism and cross-kingdom communication were conceived to refer to the interspecific interactions apart from the classic ecological relations, describing the communication at the molecular and cellular levels.”
- Table S1: how can you explain the presence in EVs of proteins such as ion channels or HSPs, or signal transduction proteins which basically have a function in the cells? Are they really from EVs or could they just originate from cell debris? This point should be clearly addressed in the discussion.
A: Table S1 shows EV markers identified in Bj-EVs. To be considered an EV marker, these proteins must be enriched in the vesicle when compared to the parental cell [2]. Moreover, they must be present in vesicles from distinct origins. Membrane and cytosolic proteins are shown to be specifically loaded during the biogenesis of the vesicles [3]. It is important to clarify that the proteins from Table S1 are already assigned as conserved and enriched in EVs, no matter the biological source. Furthermore, these proteins are consolidated as EV markers in databases collected from many proteomics studies of extracellular vesicles. For instance, Chloride Intracellular Channel Protein 1 (CLIC1) was identified in 251 proteomic studies in the last update in Vesiclepedia (http://microvesicles.org/extracellular_vesicle_markers). While HSPs were identified in more than 300 studies. Note that each protein in Table S1 has at least two references that consolidate them as markers from wide proteomic studies or gathered in reviews. For more details on HSP, please see the review by Taha et al., 2019 [4].
We included the following phrase in the discussion, Lines 346-349: “EV-markers are recurrently identified proteins enriched in certain types of EVs compared to their parental cells. The EV-markers can be either membrane or cytosolic proteins that are specifically loaded in EVs during their biogenesis.”
- Figs4-5: why not perform the same experiment with raw venom to evaluate if Evs of the venom are not internalized as well, to place EVs in a more physiological situation?
A: We appreciate the suggestion. In this first work, we aimed to verify the interaction of EVs themselves with cells without overlapping the effect of major toxins. It is worth emphasizing that viperid crude venoms, including B. jararaca, are highly cytotoxic. This will be considered in further studies since we have shown in this work the interaction of the EVs alone, but we need to be careful with the protocol due to the cytotoxic effect of crude venom, considering the EV:crude venom ratio.
- Discussion: the authors propose that the function of EVs is linked to the processing of toxic proteins, i.e. their maturation to a final functional form, rather like hormones. However, the concept of toxicodynamic can be reinterpreted in the light of this work. It is indeed imagined that a venom contains protein toxins that are released during envenomation and transported as such in the internal environment of the target. This study suggests that toxins are transported to their physiological targets via EVs. This concept, in my opinion, needs to be further clarified to shed light on the functionality of EVs. Is the benefit of venom toxins being present in EVs to be protected from the target's resident proteases at the time of envenomation?
A: We hypothesized that EVs might participate in the maturation of toxins in the venom gland, based both on the physiology of the viperid glands (which matures their toxins after secretion, with no current elucidated mechanism) and on the enrichment of processing enzymes in EVs, as demonstrated in the proteomic characterization (table 2). As venom evolved to be multifunctional, having several molecular targets simultaneously, the processing hypothesis does not exclude EVs from having other effects on the envenomed organism. We meant that the EVs containing their minor toxins will reach their cellular targets. Not necessarily the major toxins will depend on EVs to be distributed. However, one of the pharmacological actions of adenosine (a product generated by 5NTD, which is the most abundant protein in vesicles) is to increase vascular permeability. In this sense, EVs could indirectly help the major venom toxins to spread.
At this point, we cannot discard the benefit of toxins linked to EVs of being protected from proteases. In fact, this point can add another possible function to the EVs and deserves to be further investigated.
We complemented the discussion section accordingly.
- Schneider, E.; Winzer, R.; Rissiek, A.; Ricklefs, I.; Meyer-Schwesinger, C.; Ricklefs, F.L.; Bauche, A.; Behrends, J.; Reimer, R.; Brenna, S.; et al. CD73-mediated adenosine production by CD8 T cell-derived extracellular vesicles constitutes an intrinsic mechanism of immune suppression. Nat. Commun. 2021, 12, 1–14, doi:10.1038/s41467-021-26134-w.
- Jeppesen, D.K.; Fenix, A.M.; Franklin, J.L.; Higginbotham, J.N.; Zhang, Q.; Zimmerman, L.J.; Liebler, D.C.; Ping, J.; Liu, Q.; Evans, R.; et al. Reassessment of Exosome Composition. Cell 2019, 177, 428-445.e18, doi:10.1016/j.cell.2019.02.029.
- Van Niel, G.; D’Angelo, G.; Raposo, G. Shedding light on the cell biology of extracellular vesicles. Nat. Rev. Mol. Cell Biol. 2018, 19, 213–228, doi:10.1038/nrm.2017.125.
- Taha, E.A.; Ono, K.; Eguchi, T. Roles of extracellular HSPs as biomarkers in immune surveillance and immune evasion. Int. J. Mol. Sci. 2019, 20, doi:10.3390/ijms20184588.
Reviewer 3 Report
Authors performed the characterization of Extracellular vesicles (EVs) from Bothrops jararaca venom in term of their structure, protein composition and interaction with mammalian cells .
The manuscript is well-written, and application of materials and methods are appropriate.
Author Response
Reviewer3
Authors performed the characterization of Extracellular vesicles (EVs) from Bothrops jararaca venom in term of their structure, protein composition and interaction with mammalian cells.
The manuscript is well-written, and application of materials and methods are appropriate.
A: We thank the reviewer for the comments.
Round 2
Reviewer 2 Report
The authors of this study have answered the questions that were addressed to them in the first Round. I think the article is in a form that allows it to be considered for publication. There are 159 references in the bibliography, which seems like too many and the authors should be more discerning in choosing the articles they cite in this paper. Other than that, the paper is a good study, and will certainly be cited itself.